# A Cross-Sectional Study on Physical Activity and Psychological Distress in Adults with Asthma

**DOI:** 10.3390/healthcare10122469

**Published:** 2022-12-07

**Authors:** Ángel Denche-Zamorano, Javier Urbano-Mairena, Raquel Pastor-Cisneros, Laura Muñoz-Bermejo, Sabina Barrios-Fernandez, Miguel Ángel Garcia-Gordillo, Alexis Colmenarez-Mendoza, Joan Guerra-Bustamante, María Mendoza-Muñoz

**Affiliations:** 1Promoting a Healthy Society Research Group (PHeSO), Faculty of Sport Sciences, University of Extremadura, 10003 Caceres, Spain; 2Social Impact and Innovation in Health (InHEALTH), University of Extremadura, 10003 Caceres, Spain; 3Occupation, Participation, Sustainability and Quality of Life (Ability Research Group), Nursing and Occupational Therapy College, University of Extremadura, 10003 Caceres, Spain; 4Universidad Autónoma de Chile, Talca 3467987, Chile; 5Departamento de Administración y Economía, Universidad de La Frontera, Temuco 4811230, Chile; 6Area of Personality, Evaluation and Psychological Treatment, Department of Psychology and Anthropology, Faculty of Nursing and Occupational Therapy, University of Extremadura, 10003 Caceres, Spain; 7Research Group on Physical and Health Literacy and Health-Related Quality of Life (PHYQOL), Faculty of Sport Sciences, University of Extremadura, 10003 Caceres, Spain; 8Departamento de Desporto e Saúde, Escola de Saúde e Desenvolvimento Humano, Universidade de Évora, 7004-516 Évora, Portugal

**Keywords:** depression, health, mental health, perceived social support, psychology, self-perceived health

## Abstract

Physical Activity (PA) could contribute to decreasing psychological distress and improving Self-Perceived Health (SPH) in adults with Asthma. The main objective of this study was to analyse the associations between the Physical Activity Level (PAL) Perceived Social Support (PSS) and Mental Health, using the Goldberg General Health Questionnaire (GHQ-12), and SPH in the adult population with Asthma. This descriptive cross-sectional study is based on data from the 2017 Spanish National Health Survey, including 1040 participants with Asthma in the study. The Kruskal–Wallis test was performed to study the hypothetical differences between the PAL and the different variables derived from the GHQ-12. In addition, correlations between the variables generated and the items of the GHQ-12, together with the PAL and the Duke-UNC-11, were analysed using Spearman’s rho correlation coefficients. Inverse correlations were found between Mental Health and PSS (rho: −0.351) and between Mental Health and PAL (rho: −0.209), as well as in the rest of the GHQ-12 items: successful coping (rho: −0.197), self-esteem (rho: −0.193) and stress (rho: −0.145). The more active subjects had better SPH. Therefore, the research showed how higher PAL and positive SPH are related to lower psychological distress in adults with Asthma.

## 1. Introduction

Asthma is a chronic inflammatory disease of the airways, involving numerous cells and cellular elements. Pathophysiologically, it is characterised by reversible bronchial obstruction and airway hyperresponsiveness, and clinically by recurrent episodes of coughing, dyspnoea and wheezing [1]. Asthma prevalence increased significantly in the 20th century and is now estimated to be around 10% in Europe [2]. According to the World Health Organisation (WHO), there are more than 300 million Asthma patients in the world [3]. Under-utilisation of prescribed therapies has been shown to require more emergency treatment, with a considerable cost to health systems [4]. In line with this fact, it can be observed that in low- and middle-income countries, Asthma is responsible for more than 80% of deaths [3]. 

Current Asthma treatment guidelines mainly include medication management with anti-inflammatory drugs and long-acting bronchodilators [5]. In this sense, it has been shown that Physical Activity (PA) can be considered a protective factor against the development of Asthma [6]. There is evidence highlighting the importance of PA in improving cardiopulmonary fitness, Asthma symptoms and quality of life in asthmatics [7]. Tests consistently agree that PA low levels have been associated with increased Asthma symptoms, healthcare utilization and decreased quality of life [8,9,10,11]. Currently, the practice of PA in people with Asthma is recommended, due to studies reporting that performing PA in people with Asthma is safe and produces benefits for respiratory health [12]. However, previous studies have shown that people with Asthma are less likely to engage in PA and less likely to engage in more intense exercise than people without Asthma [13,14,15], i.e., people with Asthma engage in less activity compared to controls [16]. Based on the above, it is suggested that both training and PA high levels play an essential role in the development and severity of Asthma.

Several epidemiological studies have identified that adults with Asthma are more likely to experience psychological comorbidities, in particular anxiety and depression, compared to the general population. Specifically, Adams et al. [17] showed that psychological distress was more common in people with Asthma than in those without Asthma (17.9% vs. 12.2%), as was the risk of anxiety or depression (40.5% vs. 31.2%), [17]. In the United States, out of a total of 7% of adults with Asthma, the severe psychological distress prevalence was 7.5% [18]. Focusing on these two factors, various research link PA and mental health. Physical exercise is related to mental disorders symptom reduction [19], psychological well-being improvement [20], cognitive abilities development [21] and stress reduction [22]. This suggests that those who are more physically active have a lower risk of suffering from mental disorders than those who are more inactive [23]. Therefore, PA helps the population to improve their quality of life, resulting in improved physical health and psychological well-being [24]. In addition, PA could be used as a measure for mild and moderate depressive symptoms [25]. Both aerobic training and strength and resistance training seem to be effective in treating symptoms of depression and anxiety [25].

In 2020, the COVID-19 pandemic began. During the pandemic period, an episode of social isolation occurred, which could have negative effects on mental health [26]. Due to these circumstances, mental health started to become more important in our society. This fact mainly reports that the pandemic had a negative impact on society [27]. Social isolation caused anxiety, depression and stress around the world [28]; in addition, due to the reduction of people’s PA, worsened psychological distress was reflected among the population [29]. This shows a direct positive relationship between mental health and PA, meaning those more physically active reported better mental health [30]. Perceived social support (PSS) refers to the subjective assessment of how people perceive their family and friends’ concern and offer of psychological and general support at times when a given situation requires it [31]. There is evidence of a significant association between mental health and PSS [32]. High levels of PSS are associated with a lower risk of mental disorders such as depression [33]. Not only can a low PSS be related to a higher risk of mental disorder, but it has also been shown that those with a high PSS also have better sleep quality [31]. Therefore, the PSS, like PA, could be an indicator of psychological well-being that helps to prevent mental disorders such as depression [34].

Therefore, this research aims to study the correlation between mental health, Physical Activity Level (PAL), PSS and Self-Perceived level of Health (SPH) in Spanish adults with diagnosed Asthma. It is hypothesised that higher PAL, PSS and SPH are related to lower indicators of psychological distress.

## 2. Materials and Methods

### 2.1. Design

The design followed in this research was that of a descriptive cross-sectional study, based on the data included in the public files from the Spanish National Health Survey 2017 (ENSE 2017), adult questionnaire [35]. The ENSE is a survey promoted by the Spanish Ministry of Health, Consumer Affairs and Social Welfare (MSCBS), together with the Spanish National Statistics Institute (INE), to find out the health status of the Spanish resident population over 15 years of age. These surveys are carried out every five years, and the results are published in the year following the survey. ENSE 2017 was the last ENSE conducted in Spain before the COVID-19 pandemic, providing relevant information on the state of health of the Spanish population before the pandemic situation was experienced globally, allowing comparison with the results of surveys conducted during the pandemic and post-pandemic. The surveys were conducted by MSCBS-trained and accredited interviewers between October 2016 and October 2017, with data published in the summer of 2018.

### 2.2. Participants

The ENSE 2017 had a sample of 23,089 participants, selected using a stratified three-phase random sampling system [36]. This system, as well as the sample calculation, was fully described in the survey methodology [35,37]. Upon selection, participants were informed of their inclusion in the ENSE 2017, the nature of the survey, the confidential processing of the data and their anonymous publication, and they had to provide consent to participate in the survey. 

#### 2.2.1. Exclusion Criteria

In this research, data were taken from all participants in the ENSE 2017, and the following participants were excluded: those aged 70 years and older (5312), and those who reported no Asthma in item Q.25a.10 (16,726), participants with no response to any of the questions corresponding to the Goldberg General Health Questionnaire (GHQ-12), Q.47.1–Q.47.12 [35,38] and participants with no data on some of the questions corresponding to the International Physical Activity Questionnaire (IPAQ), Q.112–Q.117 [39]. 

#### 2.2.2. Inclusion Criteria

As inclusion criteria, participants had to be younger than 70 years (those older than 70 years were not questioned about their physical activity in the ENSE 2017), declare to be asthmatics, and submit all answers to the corresponding questions of the GHQ-12 and the IPAQ questionnaire. The final sample was composed of 1040 participants.

### 2.3. Variables

The data were extracted from the public files of the ENSE 2017, processing the following variables:

**Sex**: Male or female. 

**Age**: Years. 

**Self-Perceived Health (SPH):** This was obtained from the answers given to item Q.21 (“In the last twelve months, would you say that state of health has been very good, good, fair, poor, bad, or very bad?”). In this research, we considered Negative SPH as “Fair/Bad/Very bad” responses and Positive SPH as “Good/Very Good” responses.

**Asthma Status:** From item “Q.25a. 10” onwards (“Have you ever been affected by Asthma?” Yes or No).

**Social Class**: Extracted from the variable “Clase_PR” from the ENSE 2017, this variable grouped participants into 6 levels, according to their social class, based on their professional occupation, ranging from Class I (directors and managers with 10 or more employees) to Class VI (unskilled workers). This variable is further defined in the “Appendix: Social Class” of the ENSE 2017 methodology [37].

**Mental health**: For the construction of this variable, the GHQ-12 was used, a multi-factor scale that assesses general mental health and three factors of mental health (successful coping, self-esteem and stress) [40,41], based on 12 questions with answers that can take values between 0 and 3, where 0 is the best mental health and 3 is the worst. The overall score for mental health is constructed by adding the scores of the 12 items, with 0 being the best mental health and 36 the worst. In the Spanish population, people with 12 or more scores on the GHQ-12 are considered to have some form of psychological distress [38,40,41,42]. The GHQ-12 has a high internal consistency in the Spanish population for assessing psychological distress and short-term changes in mental health (α = 0.86). In the ENSE 2017, the items corresponding to the GHQ-12 were Q.47.1–Q47.12. Three factors of mental health were analysed, according to the GHQ-12 [38].

**Successful Coping**: It was calculated as the sum of the responses given to items Q.47.1, Q.47.3, Q.47.4, Q.47.7, Q.47.8 and Q.47.12. The highest score could reach 18 points, being the worst condition of successful coping, with 0 being the best. In the Spanish population, this factor has a validity with an α of 0.82. [40]

**Self-esteem**: This was calculated as the sum of the answers given to items Q.47.6, Q.47.9, Q.47.10, and Q.47.11. The highest score that this factor could take was 12, being the worst self-esteem, and 0 the best. In the Spanish population, this factor has a validity with an α of 0.70 [40].

**Stress:** It was calculated as the sum of the responses to items Q.47.2, Q.47.5, and Q.47.9. The highest score that this factor could take was 9, being the worst self-esteem, and 0 was the best. In the Spanish population, this factor has a validity with an α of 0.75 [40].

**Perceived Social Support (PSS)**: This variable was calculated with the answers given to the Duke-UNC-11 Functional Social Support Questionnaire, items Q.130.1–Q.130.11 of the ENSE 2017. This questionnaire assesses participants’ perception of the social support they have in their daily life. There are 11 items; answers take values between 1 and 5, where 1 is the least support and 5 is the best. Thus, the maximum possible score is 55 points, the highest PSS. In the Spanish population, a PSS lower than 32 indicates a low PSS. The internal consistency of this questionnaire in the Spanish population is excellent, with an α of 0.90 [40,43].

**Physical Activity Level (PAL)**: The population was grouped into four levels of physical activity: inactive, walker, active or very active. For this purpose, a physical activity index (PAI) was constructed, taking into account the answers given to items Q.112–Q.117, corresponding to the IPAQ short, Spanish version [39]. This PAI was based on an adaptation by Denche et al. [44], described and used in multiple previous publications, based on the PAI of Ness et al. [45]. The PAI was constructed by applying a series of factors to the answers given by participants to questions related to the intensity, frequency and duration of physical activity usually performed in a week. The factors applied were previously described by Nes et al. [45].

The PAI could take values between 0 and 67.5, with 67.5 being the highest physical activity and 0 being the lowest [44]. To obtain a score of 0, participants were required not to engage in any moderate or vigorous physical activity per week. Among the participants with PAI equal to 0, participants who reported not walking were considered “Inactive” (Q.117 “Now think about how much time you spent walking in the last 7 days”, those who answered “No day more than 10 min at a time” or “less than 1 to 7 days”), while those who reported walking more than one day a week, more than 10 min in a row, were considered “Walkers” (Q.117 “Now think about how much time you spent walking in the last 7 days”, answered “Each day more than 10 min at a time” or “from 1 to 7 days”).

Among participants with a PAI greater than 0, participants with PAI scores between 1 and 30 were considered “Active”, while participants with PAI scores greater than 30 were considered “Very active”.

### 2.4. Statistical Analysis

The normality of the data of the variables to be analysed was studied using the Kolmogorov–Smirnov test. The sample was characterised by performing a descriptive analysis, presenting the median values and the interquartile range, complemented with the mean values and standard deviation of the continuous variables (Age, PAI, Mental health, Successful coping, Self-esteem, Stress and PSS), and the absolute and relative frequencies presented by the categorical variables (PAL and SPH). 

Non-parametric statistical tests were performed to analyse possible inter-group or baseline differences (Mann–Whitney U test, Kruskal–Wallis test) for continuous variables and to assess possible dependence relationships (Chi-square test) and differences in proportions between groups (pairwise z-test for independent proportions) for categorical variables. A Spearman correlation analysis was performed, using the Bonferroni correction, interpreted according to Cohen’s classification.

Multiple binary logistic regression was performed, taking SPH as the dependent variable and PAL, Sex, Age, PSS, Social Class and BMI as independent variables. Finally, linear regressions were performed to predict the scores of the variable’s mental health, successful coping, self-esteem and stress, taking these as dependent variables and PAL, Sex, BMI, Social Class, Age and PSS as independent variables. For all analyses, a level of less than 0.05 was considered statistically significant. IBM SPSS Statistical v.25 software was used for all analyses in this study.

## 3. Results

Table 1 shows the descriptive analysis used to characterise the sample, finding that the median PAI in the general population with Asthma was zero, as in men and women, although there were significant differences between sexes (*p* < 0.001). In contrast to the median, men had a higher mean PAI than women (14.4 vs. 9.1). In that sense, sex-dependent relationships were found with PAL (*p* < 0.001), with higher proportions of active (31.4%) and very active (16.1%) men than women (24.2% and 8.8%, respectively), with significant differences between proportions (*p* < 0.05). Median scores on the GHQ-12 were 10 points in the general population and both sexes. Significant differences were found between men and women (*p* < 0.05), with a lower mean score in men than in women (10.6 vs. 11.4). The median PSS was 49 in the general population with Asthma, with no differences between sexes (*p* = 0.340). PSS and sex showed dependency relationships (*p* < 0.05), with women having a higher proportion of negative PSS than men (44.9% vs. 36.9%), with significant differences between proportions (*p* < 0.05).

Table 2 shows the SPH of the asthmatic population as a function of PAL, as well as the dependency relationships found between the two variables (*p* < 0.001). The results showed the highest levels of positive self-perceived health (76.4%) in the very active group, while the inactive group had the lowest proportions (40.4%), with significant differences between these proportions (*p* < 0.05). Figure 1 shows negative SPH prevalence, according to the participants’ PAL.

The highest mental health scores on the GHQ-12 were found in the inactive group (13.8) in both men and women (14.1 vs. 13.7), while the lowest scores were found in the very active group (9.3): very active men (9.2) and very active women (9.5). Significant differences were found in the median baseline scores obtained by each PAL group in the general population and both sexes, both in the GHQ-12 and in its subdimensions (*p* < 0.001) (Appendix A).

Inverse correlations were found between the PAL and the GHQ-12 (rho: −0.209, *p* < 0.001). These inverse correlations also held for the different sub-dimensions of the GHQ-12: Successful Coping (rho: −0.197, *p* < 0.001), Self-esteem (rho: −0.193, *p* < 0.001) and Stress (rho: −0.145, *p* < 0.001). The same direction of the correlations was maintained for each of the items of the GHQ-12, with statistically significant values (Table 3).

Similarly, inverse correlations were found between PSS and mental health, according to the GHQ-12 questionnaire (rho: −0.351. *p* < 0.001) and in its different subscales: Successful coping (rho:−0.266. *p* < 0.001), Self-esteem (rho: −0.343. *p* < 0.001) and Stress (rho: −0.334. *p* < 0.001). These inverse correlations were maintained in each of the subdimensions of the GHQ-12, as well as in each of its items, being statistically significant values (Table 4).

Finally, a multiple binary logistic regression analysis was performed for risk factors for negative SPH, which found that men, active and very active individuals, and those with higher PSS and social class had reduced risks of negative SPH. This model explained 27.1% (R^2^ Nagelkerke) of the variance in the SPH (Table 5).

Finally, the linear regression model presented a coefficient of determination:R2 = 17.7%, positively explained by Mental Health (Constant: β = 20.176, t = 11.959, *p* < 0.001; PSS: β = −0.237, t = −10.187, *p* < 0.001; PAL: β = −0.940, t = −4.796, *p* < 0.001; BMI: β = −0.075, t = 2.220, *p* = 0.027; Social Class: β = 0.275, t = 2.557, *p* = 0.011; Age: β = 0.030, t = 2.433, *p* = 0.015).R2 = 12.0%, positively explained by Successful Coping (Constant: β = 9.901, t = 16.200, *p* < 0.001; PSS: β = −0.070, t = −7.643, *p* < 0.001; PAL: β = −0.472, t = −6.138, *p* < 0.001; BMI: β = 0.040, t = 3.087, *p* = 0.002).R2 = 17.2%, positively explained by Self-esteem (Constant: β = 6.739, t = 9.440, *p* < 0.001; PSS: β = −0.108, t = −9.809, *p* < 0.001; PAL: β = −0.392, t = −4.271, *p* < 0.001; Age: β = 0.022, t = 4.001, *p* < 0.001; Social Class: β = 0.191, t = 3.798, *p* < 0.001).R2 = 13.9%, positively explained by Stress (Constant: β = 6.258, t = 8.638, *p* < 0.001; PSS: β = −0.099, t = −10.238, *p* < 0.001; PAL: β = −0.185, t = −2.249, *p* = 0.025; Age: β = 0.010, t = 1.879, *p* = 0.061; Sex: β = 0.388, t = 2.750, *p* = 0.006; BMI: β = 0.030, t = 2.750, *p* = 0.036).

## 4. Discussion

### 4.1. Main Findings

This research aimed to study the relationship between mental health, PAL, PSS and SPH in Spanish adults with Asthma, and the main finding was that elevated PAL and SPH levels were associated with improved mental health and thus reduced psychological distress in adults with Asthma.

In terms of PAL, significant associations were found between both sexes (*p* < 0.001). The results showed that men were more physically active than women (14.4 vs. 9.1) and women were more inactive than men (15.9% vs. 12.6%), which is in line with previous research on the amount of PA performed by men and women in different age ranges [22,46,47,48,49]. In terms of who were the most active subjects, men again scored higher than women (16.1% vs. 8.8%), similar to previous studies [48,49].

Regarding SPH, being physically inactive or doing insufficient PA could be found to be related to poor mental health and negative SPH. As far as our research is concerned, men performed better than women, with significant differences (*p* < 0.010) between both sexes, as well as a dependence relationship between SPH and male sex (B: −0.296), in line with a previous study where the trend in females was characterised by a worse SPH [50]. Significant differences were found between positive (44.9) and negative (55.1) SPH in women (*p* < 0.001). The study revealed that more inactive participants had a greater negative impact on SPH (inactive 59.6% vs. very active 23.6%), which is in line with the results of previous research [51,52].

Regarding PSS, no significant differences were found (47.3 vs. 47.6) between the sexes in contrast to other research [53]. This score is above 32 points, which would indicate high social support [40]. However, despite no significant differences (*p* < 0.340), women were slightly superior, in line with studies where women tend to score higher than men [37]. In addition, inverse correlations were found between the Goldberg Mental Health Questionnaire (GHQ-12) (rho: −0.287, *p* < 0.001) and the PSS and its subscales, successful coping (rho: −0.173, *p* < 0.001), self-esteem (rho: −0.281, *p* < 0.001) and stress (rho: −0.292, *p* < 0.001), as well as in the different items of the Duke-UNC-11 questionnaire. These results would indicate that a high PSS would mean less stress, anxiety and depression, as indicated by previous studies [22]. 

Regarding mental health outcomes, a score of 11.9 was obtained, with 36 being the worst possible score on the GHQ-12 [41]. A small correlation was observed between PAL and mental health (rho: −0.209, *p* < 0.001), which could indicate that the higher PAL, the lower the psychological distress and risk of mental disorders [54,55]. Therefore, this could suggest that the higher the PAL, the lower the scores on the GHQ-12, denoting better psychological well-being [56] and less psychological distress [57]. The highest score on the GHQ-12 was obtained by the inactive group, while the lowest score was obtained by the very active group (13.8 vs. 9.1), so more inactive people would have greater psychological distress and a higher risk of suffering from a mental disorder. Therefore, our research shows that PA could help to improve mental health, as mentioned in numerous studies [55,58,59], even reducing the risk of mental illness [60]. However, Soon Kim et al. [61], in their study on PA and mental health, suggest that there is a range of optimal PA for the prevention of mental illness, where the most active subjects may not always have the best mental health; this may be at odds with the results obtained in this study. 

Likewise, the inverse correlation was also maintained in the other subscales of the questionnaire: successful coping (rho: −0.197, *p* < 0.001), self-esteem (rho: −0.193, *p* < 0.001) and stress (rho: −0.145, *p* < 0.001), as well as in the rest of the items of the GHQ-12. Looking at the correlations, it can be seen that both successful coping and self-esteem showed the greatest benefits. Therefore, these results are in line with studies that indicate that PA could help to prevent mental disorders such as depression, anxiety or other mental disorders [22,55,60].

### 4.2. Practical Applications

The importance of this study lies in the analysis of the associations between mental health, PAL, PPS and SPH in the Spanish adult population with Asthma, during the last period before the COVID-19 pandemic, serving as a framework for future research examining post-pandemic periods, as the ENSE is addressed every 5 years.

The value of this research is the finding that the higher PAL and positive SPH, the lower the scores on the GHQ-12, which could show less psychological distress in people with Asthma and serve as a reference for future research to quantify the optimal amount of PA to improve the psychological well-being of people with Asthma, improving physical and mental health. 

It would be interesting for PA professionals, health professionals and political institutions to work together to promote and create programmes of moderate and vigorous physical activity, so that the asthmatic population practices PA several days a week, to reduce the inactivity of this population and to improve their physical and mental health. Increasing the PA time of individuals with Asthma would improve their PSS, allowing them to reduce stress, anxiety and other psychological disorders. However, further research is needed to establish cause-effect relationships and to assess the proper dose and the correct prescription of this kind of programs.

### 4.3. Limitations

Due to the cross-sectional design of this study, one of its limitations is the impossibility of establishing cause-effect relationships; thus, longitudinal studies that allow for this should be a future line of research. Measuring PA using questionnaires is a limitation, as it is not possible to quantify PA more objectively. Measuring this using electronic devices such as physical activity wristbands or pedometers is a more effective way of measuring PA, helping to gather more information on optimal PA time ranges for this population, which may become a future line of research. In addition, no different questionnaires were set up to further specify the amount of PA, so there was no distinction between participants who engaged in moderate or vigorous PA or walked more frequently. In this sense, it would be useful to be able to measure the number of hours and frequencies of PA to understand the relationships between variables. Although some confounding factors (age, sex, BMI and social class) were considered, other potential sources of bias could not be addressed, as they would be almost endless. Other variables could be considered as future lines (marital status, smoking, alcohol consumption), but they are outside the scope of this work.

## 5. Conclusions

PA and SPH are related to good mental health in the adult Asthma population. Therefore, adults with Asthma who perform PA and have a positive SPH have less psychological distress, relative to the GHQ-12 data. Thus, PA could influence the reduction of psychological distress in adults with Asthma.

## Figures and Tables

**Figure 1 healthcare-10-02469-f001:**
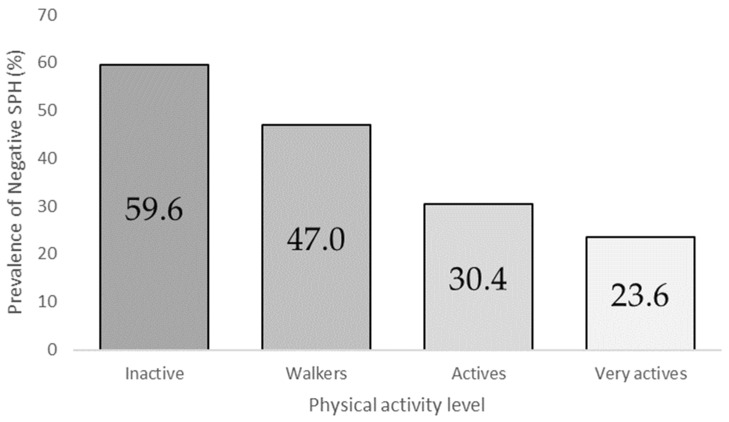
Negative SPH prevalence according to the Physical Activity Level (PAL).

**Table 1 healthcare-10-02469-t001:** Descriptive analysis of the variables studied (PAL, SPH, PSS and mental health dimensions) in Spanish adults with Asthma, according to the ENSE 2017.

Variables	Total *n* = 1040	Men *n* = 436	Women *n* = 604	
Age (Years)				*p*
Median (IQR)	43 (23)	42 (21)	44 (24)	0.217
Mean (SD)	43.1 (14.7)	42.5 (14.3)	43.5 (14.9)	
PAI				*p*
Median (IQR)	0 (22)	0 (30)	0 (15)	<0.001
Mean (SD)	11.3 (17.2)	14.4 (18.1)	9.1 (15.7)	
Mental health				*p*
Median (IQR)	10 (6)	10 (5)	10 (6)	0.032
Mean (SD)	11.1 (5.6)	10.6 (5.3)	11.4 (5.8)	
Successful coping				*p*
Median (IQR)	6 (1)	6 (0)	6 (1)	0.353
Mean (SD)	6.5 (2.2)	6.4 (2.0)	6.6 (2.3)	
Self-esteem				*p*
Median (IQR)	2 (4)	1 (4)	2 (4)	0.028
Mean (SD)	2.4 (2.7)	2.6 (2.5)	2.6 (2.7)	
Stress				*p*
Median (IQR)	3 (3)	3 (3)	3 (4)	0.009
Mean (SD)	2.9 (2.2)	2.7 (2.2)	3.1 (2.4)	
Perceived social support	Total *n* = 1010	Men *n* = 426	Women *n* = 584	*p*
Median (IQR)	49 (9)	49 (13)	50 (9)	0.340
Mean (SD)	47.5 (7.2)	47.3 (7.3)	47.6 (7.1)	
SPH *n* (%)	Total *n* = 1040	Men *n* = 436	Women *n* = 604	*p**
Negative	432 (41.5)	161 (36.9)	271 (44.9) *	0.010
Positive	608 (58.5)	275 (63.1)	333 (55.1) *	
PAL *n* (%)				
Inactive	151 (14.5%)	55 (12.6%)	96 (15.9%)	<0.001
Walkers	483 (46.4%)	174 (39.9%)	309 (51.2%) *	
Active	283 (27.2%)	137 (31.4%)	146 (24.2%) *	
Very active	123 (11.8%)	70 (16.1%)	53 (8.8%) *	
Social Class *n* (%)				
I	115 (11.3%)	64 (14.9%)	51 (8.7%) *	<0.001
II	86 (8.5%)	34 (7.9%)	52 (8.9%)	
III	192 (18.9%)	80 (18.6%)	112 (19.1%)	
IV	142 (14.0%)	66 (15.3%)	76 (13.0%)	
V	339 (33.4%)	136 (31.6%)	203 (34.6%)	
VI	142 (14.0%)	50 (11.6%)	92 (15.7%)	

*n* (participants); % (percentage); IQR (Interquartile range); SD (Standard deviation); GHQ-12 (Goldberg’s General Health Questionnaire. Scores between 0 and 36: 0, the best mental health; 36, the worst mental health); Successful Coping (Scores from 0 to 18: 0, the best coping; 18, the worst coping); Self-esteem (Scores 0 to 9: 0, the best self-esteem; 9, the worst self-esteem); Stress (Scores 0 to 9: 0, no stress; 9, very stressed); PAL (Physical Activity Level); Inactive (PAI = 0; declare not going for a walk, no day a week, more than 10 min at a time). Walkers (PAI = 0; report walking, at least one day a week, more than 10 min at a time). Active (PAI = 1–30); Very active (PAI = +30); PAI (Physical Activity Index: Scores between 0 and 67.5); *p* (*p*-value from U-Mann-Whitney test); *p** (*p*-value from chi-square test); * (Significant differences between sex ratios, *p* < 0.05 in pairwise z-test); SPH (Self-perceived health).

**Table 2 healthcare-10-02469-t002:** Associations between the Physical Activity Level (PAL)and subscales from the mental health in Spanish adults with Asthma, according to the ENSE 2017.

Variables	Total *n* = 1006			Men *n* = 572			Women *n* = 434		
**Mental health**									
PAL	m (sd)	mdn (IQR)	*p*	m (sd)	mdn (IQR)	*p*	m (sd)	mdn (IQR)	*p*
Inactive	13.8 (7.2)	12 (8)	<0.001	14.1 (7.4)	12 (9)	<0.001	13.7 (7.1)	12 (7)	<0.001
Walker	11.2 (5.4)	10 (6)		10.3 (4.3)	10 (5)		11.7 (5.8)	11 (7)	
Active	10.2 (5.0)	9 (6)		10.2 (5.1)	9 (6)		10.1 (4.8)	9 (6)	
Very active	9.3 (4.1)	9 (6)		9.2 (4.3)	9 (5)		9.5 (4.3)	9 (6)	
**Successful coping**									
PAL	m (sd)	mdn (IQR)	*p*	m (sd)	mdn (IQR)	*p*	m (sd)	mdn (IQR)	*p*
Inactive	7.7 (3.0)	6 (3)	<0.001	7.8 (2.0)	6 (3)	<0.001	7.6 (3.0)	6 (3)	<0.001
Walker	6.5 (3.9)	6 (0)		6.3 (2.3)	6 (0)		6.6 (2.2)	6 (1)	
Active	6.2 (3.3)	6 (0)		6.2 (1.8)	6 (0)		6.1 (1.8)	6 (1)	
Very active	5.8 (1.7)	6 (1)		5.8 (1.8)	6 (0)		5.7 (1.7)	6 (1)	
**Self-esteem**									
PAL	m (sd)	mdn (IQR)	*p*	m (sd)	mdn (IQR)	*p*	m (sd)	mdn (IQR)	*p*
Inactive	3.5 (3.3)	4 (4)	<0.001	3.8 (3.5)	3 (5)	<0.001	3.4 (3.1)	3 (3)	<0.001
Walker	2.5 (2.6)	2 (4)		2.1 (2.3)	2 (3)		2.8 (2.8)	2 (4)	
Active	2.0 (2.4)	1 (4)		2.1 (2.4)	1 (4)		2.0 (2.4)	1 (3)	
Very active	1.6 (2.0)	1 (2)		1.5 (2.0)	1 (2)		1.7 (2.1)	1 (3)	
**Stress**									
PAL	m (sd)	mdn (IQR)	*p*	m (sd)	mdn (IQR)	*p*	m (sd)	mdn (IQR)	*p*
Inactive	3.7 (2.4)	3 (4)	<0.001	3.6 (2.4)	3 (3)	0.028	3.7 (2.4)	3 (4)	<0.001
Walker	3.0 (2.2)	3 (3)		2.6 (1.9)	3 (3)		3.2 (2.4)	3 (4)	
Active	2.6 (2.2)	2 (3)		2.6 (2.3)	2 (4)		2.7 (2.2)	3 (3)	
Very active	2.5 (2.2)	2 (3)		2.4 (2.1)	2 (3)		2.7 (2.4)	2 (3)	

m (mean); sd (standard deviation); mdn (median); IQR (Interquartile range); GHQ-12 (Goldberg’s General Health Questionnaire. Scores between 0 and 36: 0, the best mental health; 36, the worst mental health); Successful Coping (Scores from 0 to 18: 0, the best coping; 18, the worst coping); Self-esteem (Scores 0 to 9: 0, the best self-esteem; 9, the worst self-esteem); Stress (Scores 0 to 9: 0, no stress; 9, very stressed); PAL (Physical Activity Level: Inactive (PAI = 0; declare not going for a walk, no day a week, more than 10 min at a time). Walkers (PAI = 0; report walking, at least one day a week, more than 10 min at a time). Active (PAI = 1–30); Very active (PAI = +30); PAI (Physical Activity Index: Scores between 0 and 67.5); *p* (*p*-value from Kruskal-Wallis test).

**Table 3 healthcare-10-02469-t003:** Correlation between the Physical Activity Level (PAL) and mental health in Spanish adults with Asthma, according to the ENSE2017.

Target Variable	Rho	*p*
Mental Health	−0.209	<0.001
Successful Coping	−0.197	<0.001
Self-esteem	−0.193	<0.001
Stress	−0.145	<0.001
1. Have you been able to concentrate well on what you are doing?	−0.117	<0.001
2. Have your worries caused you to lose sleep?	−0.102	0.001
3. Do you feel that you were playing a useful role in life?	−0.139	<0.001
4. Do you feel able to make decisions?	−0.142	<0.001
5. Have you felt constantly overwhelmed and under stress?	−0.126	<0.001
6. Have you had the feeling that you cannot overcome your difficulties?	−0.163	<0.001
7. Have you been able to enjoy your normal daily activities?	−0.177	<0.001
8. Have you been able to cope adequately with your problems?	−0.205	<0.001
9. Have you felt unhappy or depressed?	−0.149	<0.001
10. Have you lost confidence in yourself?	−0.187	<0.001
11. Have you thought of yourself as a worthless person?	−0.187	<0.001
12. Do you feel reasonably happy considering all the circumstances?	−0.190	<0.001

GHQ-12 (Goldberg’s General Health Questionnaire. Scores between 0 and 36: 0, the best mental health; 36, the worst mental health); Successful Coping (Scores from 0 to 18: 0, the best coping; 18, the worst coping); Self-esteem (Scores 0 to 9: 0, the best self-esteem; 9, the worst self-esteem); Stress (Scores 0 to 9: 0, no stress; 9, very stressed); PAL (Physical Activity Level: Inactive (PAI = 0; declare not going for a walk, no day a week more than 10 min at a time). Walkers (PAI = 0; report walking, at least one day a week, more than 10 min at a time). Active (PAI = 1–30); Very active (PAI = +30); PAI (Physical Activity Index: Scores between 0 and 67.5); Rho (Spearman’s correlation coefficients with the Bonferroni correction factor having *p* = 0.003); *p* (*p*-value).

**Table 4 healthcare-10-02469-t004:** Correlation between Pelf Perceived Support (SPP) and mental health in Spanish adults with Asthma, according to the ENSE2017.

Target Variable	Correlations	*p*
Mental Health	−0.287	<0.001
Successful Coping	−0.173	<0.001
Self-esteem	−0.281	<0.001
Stress	−0.292	<0.001
1. Have you been able to concentrate well on what you are doing?	−0.127	<0.001
2. Have your worries caused you to lose sleep?	−0.207	<0.001
3. Do you feel that you are playing a useful role in life?	−0.152	<0.001
4. Do you feel able to make decisions?	−0.096	0.002
5. Have you felt constantly overwhelmed and under stress?	−0.255	<0.001
6. Have you had the feeling that you cannot overcome your difficulties?	−0.228	<0.001
7. Have you been able to enjoy your normal daily activities?	−0.144	<0.001
8. Have you been able to cope adequately with your problems?	−0.143	<0.001
9. Have you felt unhappy or depressed?	−0.286	<0.001
10. Have you lost confidence in yourself?	−0.211	<0.001
11. Have you thought of yourself as a worthless person?	−0.188	<0.001
12. Do you feel reasonably happy considering all the circumstances?	−0.114	<0.001

GHQ-12 (Goldberg’s General Health Questionnaire. Scores between 0 and 36: 0, the best mental health; 36, the worst mental health); Successful Coping (Scores from 0 to 18: 0, the best coping; 18, the worst coping); Self-esteem (Scores 0 to 9: 0, the best self-esteem; 9, the worst self-esteem); Stress (Scores 0 to 9: 0, no stress; 9, very stressed); Duke-UNC-11 (Duke-UNC-11 Functional Social Support Questionnaire. Scores between 11 and 55 points); Correlations (Spearman’s correlation coefficients with the Bonferroni correction factor having *p* = 0.003); *p* (*p*-value).

**Table 5 healthcare-10-02469-t005:** Multiple binary logistic regression analysis for negative Self Perceived Health (SPH).

	B	SE	Wald	df	Sig	Exp(B)	95% C.I. for EXP(B)	
							Lower	Upper
PAL: (Inactive)			16.190	3	0.001			
Walker	−0.428	0.220	3.780	1	0.052	0.652	0.424	1.003
Active	−0.873	0.243	11.938	1	0.000	0.417	0.259	0.672
Very active	−0.924	0.311	8.844	1	0.003	0.397	0.216	0.730
Social Class: (I)			12.921	5	0.024			
II	0.083	0.353	0.055	1	0.814	1.086	0.544	2.171
III	0.564	0.284	3.930	1	0.047	1.757	1.006	3.068
IV	0.451	0.306	2.173	1	0.140	1.569	0.862	2.857
V	0.693	0.266	6.780	1	0.009	2.000	1.187	3.370
VI	0.897	0.308	8.509	1	0.004	2.453	1.342	4.484
Sex (Men)	0.275	0.154	3.193	1	0.074	1.317	0.974	1.782
Age (Years)	0.052	0.006	79.500	1	0.000	1.054	1.042	1.066
PSS (Score)	−0.041	0.011	14.564	1	0.000	0.960	0.940	0.980
IMC (Kg/m^2^)	0.028	0.015	3.301	1	0.069	1.028	0.998	1.060
Constant	−1.642	0.744	4.871	1	0.027	0.194		

B (Unstandardized beta); SE (Standard error of regression); Wald (Wald Chi-square test); Df (Degree freedom); Sig. (Statistical significance); Exp (Exponential regression); C.I. (Confidence interval).

## Data Availability

Data used were obtained from public use files, available on the Spanish Ministry of Health, Consumer Affairs, and Social Welfare website: https://www.mscbs.gob.es/estadEstudios/estadisticas/encuestaNacional/encuesta2017.htm (accessed on 1 June 2022). Additional datasets will be available upon reasonable request.

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
