# Peer review of "A Cross-Sectional Study on Physical Activity and Psychological Distress in Adults with Asthma"

_healthcare, 2022, doi:10.3390/healthcare10122469_

Round 1

Reviewer 1 Report

TITLE

- P. 1, line 2. Given the content and the objective of the article, you could consider to modify the title. 

ABSTRACT

- P. 1 line 28. You can also focus on the clinical application. 

KEYWORDS

- P.1 line 29. It could be interesting to modify the keywords selected for the study.

INTRODUCTION

After regarding the introduction section, the originality of the study is not clear to me, 

it could be interesting to go in-depth into the novelty that the study provides. 

- P. 2, line 45. It could be interesting to include information about the prevalence or  socio-demographic data of the asthma mental disturbances.

- P.1, line 55. It could be interesting to include information about the prevalence of the asthma patients who practice physical exercise.

- P.2, line 65. Do you think that the Covid-19 pandemic is a relevant aspect to mention in this study?

METHODS

- P.2, line 95. As you have commented in the methods, it could be interesting to compared the results obtained previous to COVID-19 with results obtained after COVID-19. Why don´t you do it if you have mentioned?

- P.3, line 103. It could be interesting to include a flow diagram of the included studies. 

- P.3, line 129. Could you reference the use of the single item as a validate tool for evaluating the self-perceived health?

- P.3, line 147. Could you specify which is the test that include the six items mentioned for successful coping? 

- P.4, line 150. In a similar way of the previous comment, could you clarify me if the items selected for evaluating the self-esteem are a part or are an evaluating test?

- P.4, line 154. The same as the two comments above. 

RESULTS

- P. 5, Table 1. If it is possible, it could be interesting to know more sociodemographic data of the sample.

DISCUSSION

- P.9, line 269. The aim of the study should be remembered at the beginning of the section.

CONCLUSION

- P.11, line 346. Probably, the affirmation “Therefore, adults with asthma, who perform PA and have a positive SPH, will have less psychological distress, relative to the GHQ-12 data. For that reason, PA practice would reduce psychological distress in adults with asthma.” need to be rewritten, it could not be possible to declare it analyzing the results of this study.

REFERENCES

Some references need to be reviewed, write it according to the format requested by the healthcare´s editors.

Author Response

Dear reviewer,

The response from the authors can be found in the attached file.

Reviewer 2 Report

This study examined the correlation between PAL and PSS, as well as mental health and SPH in adult with asthmatics. The manuscript is well written.

1. Please indicate any steps taken to address potential sources of bias, such as confounding factors.

2. Explain how the sample size was calculated.

3. Explain how you handled the missing data.

Author Response

(The authors gave the same response as above.)

Round 2

Reviewer 1 Report

thank you for your responses